# Tsunami heights and limits in 1945 along the Makran coast estimated from testimony gathered seven decades later in Gwadar, Pasni and Ormara

Hira Ashfaq Lodhi[1], Shoaib Ahmed[2], Haider Hasan[2]

[1]Department of Physics, NED University of Engineering & Technology, Karachi, 75270, Pakistan

[2] Department of Civil Engineering, NED University of Engineering & Technology, Karachi, 75270, Pakistan

*Correspondence to*: Hira Ashfaq Lodhi (hiralodi@neduet.edu.pk)

**Abstract.**

The towns of Pasni and Ormara were the most severely affected by the 1945 Makran tsunami. The water inundated almost a kilometre at Pasni, engulfing 80% huts of the town while at Ormara tsunami inundated two and a half kilometres washing away 60% of the huts. The plate boundary between the Arabian plate and Eurasian plate is marked by Makran Subduction Zone (MSZ). This Makran subduction zone in November 1945 was the source of a great earthquake (8.1 Mw) and an associated tsunami. Estimated death tolls, waves arrival times, the extent of inundation and runup remained vague. We summarize observations of tsunami through newspaper items, eye witness accounts and archival documents. The information gathered is reviewed and quantized where possible to get the inundation parameters in specific and impact in general along the Makran coast. The quantization of runup and inundation extents is based on a field survey or old maps.

## 1 Introduction

The recent tsunami events of 2004 Indian Ocean (Sumatra) tsunami, 2010 (Chile) and 2011 (Tohoku) Pacific Ocean tsunami have highlighted the vulnerability of coastal areas and coastal communities to such events. Credible vulnerability assessment of a coast depends upon reliable geoscientific data on past tsunami events. The data from past events is crucial as it forms the basis for numerical models that simulate tsunami and tsunami hazard assessment (Hoffmann et al., 2013) which in turn can be used for planning and mitigation and most importantly it can serve as an input for the development of tsunami early warning systems (TEWS).

The tsunami hazard of a coast is dependent upon the tsunami sources among many other parameters. The coast of Pakistan lies in close proximity of the Makran subduction zone. The historical tsunami events known in the region are sparse but have been reported by several studies (Dominey-Howes et al., 2006; Heidarzadeh et al., 2008) with the oldest one being in 325 BC (Pararas-Carayannis, 2006). The evidence of Paleo-tsunami by MSZ is debatable (Dominey-Howes et al., 2006) as the

only instrumentally recorded tsunamigenic earthquake from MSZ was in November 1945, an 8.1 Mw thrust event that occurred almost 8 km southeast of Pasni (Quittmeyer and Jacob, 1979). Another probable source of the tsunami can be landslides such as the one triggered by the 24th September 2013 inland earthquake (Hoffmann et al., 2014; Baptista et al., 2020) or potentially from the landslide on Owen ridge (Rodriguez et al., 2013).

The 1945 event being the only recorded event serves as the basis for modelling of the tsunami in the region (Rajendran et al., 2008; Heidarzadeh et al., 2008; Neetu et al., 2011) but the event itself is poorly recorded because of the aftermath of world war II and political situation of then India. We have summarized the historical accounts, eyewitness accounts and newspaper items to come up with the impact of the 1945 tsunami along the coastal cities (then towns) of Pakistan while quantizing the data where ever possible. A field survey is carried out along the three coastal cities of Gwadar, Pasni and Ormara during which inundation parameters along the three cities are identified using the landmarks reported in eyewitness accounts and newspaper items. Similar efforts have been carried out in different areas of the world over many years going back to at least the 1960 Chile tsunami. More recent ones include post tsunami field surveys of 1992 Nicaragua tsunami (Satake et al., 1993), Srilankan field survey of 2004 tsunami (Goff et al., 2006), 2010 Chile tsunami (Tsuji et al., 2010) and 2018 Sulawesi tsunami (Widiyanto et al., 2019; Mikami et al., 2019). All these surveys were carried out immediately after the tsunami event but the study presented here connects a field survey carried out recently with the tsunami event that took place approximately 70 years ago. A similar study that assesses the inundation parameters several years after the event has been conducted in Chile for the 1960 tsunami by Atwater et al., 2013. However, this technique was pioneered by Okal et al., (2002) and was applied first for the Auletian tsunami.

An effort was made by Hoffmann et al. 2013 to review and summarize historical accounts, eyewitness accounts, newspaper items and previously published work for the four countries connected by the Arabian Sea; Oman, Iran, India and Pakistan. According to the study inundation and losses were greatest along what is now the coast of Pakistan. However, the study of Hoffmann et al. 2013 did not report the runups and inundation extents or depths. A study by Okal et al., (2015), also based on field survey and eyewitness accounts quantizes the runup data along a 280 km long segment of Iranian shore. The study reports runup between 2.3–13.7 m and a time delay in the arrival of tsunami, indicating a secondary mechanism such as a landslide. Here, we report runups and inundation extents for the first time, for Gwadar, Pasni and Ormara. The findings are based on the information provided in the eyewitness accounts and newspaper items, a ground survey is conducted to locate the landmarks and come up with the runups and inundation extents along the coast of Gwadar, Pasni and Ormara.

## 2 Makran Earthquake of 1945 and Tsunami

The 1945 tsunami was a result of a thrusting event of 8.1 Mw at MSZ (Byrne et al., 1992). The earthquake was felt at Muscat, along the entire coast of Makran and many other places of now Pakistan which were far inland, e.g., Montgomery, Dadu, Dera Ismail Khan. It was widely recorded at different stations around the world (Hoffmann et al., 2013). The earthquake was followed by five recorded aftershocks (Byrne et al., 1992). The event generated a tsunami that hit the

61 countries in the north-western Indian Ocean. Fig. 1 shows the relative position of Gwadar, Pasni and Ormara relative to the

62 epicentre location of the 1945 earthquake as reported by different studies.

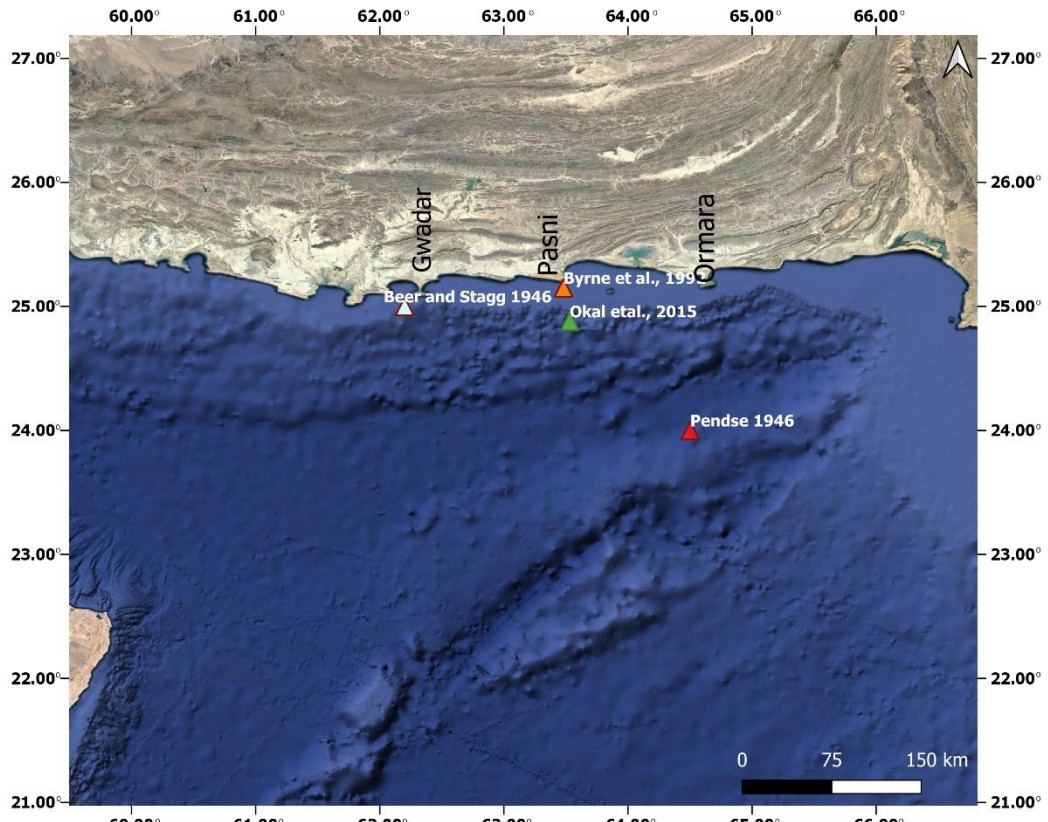

*Fig. 1 An index map showing the towns of Gwadar, Pasni and Ormara relative to Makran subduction Zone. The triangles show the epicenter for the 1945 event after different prior studies (Data plotted on © Google Satellite image).*

## 3 Impact of the 1945 Makran Tsunami

The aftermath of the 1945 Makran tsunami is not very well recorded due to the political situation of the region. The study

reports the impact of the tsunami in general and inundation parameters in specific along three coastal cities Gwadar, Pasni

and Ormara. For assessing the inundation parameters, the runup and the inundation extent, a ground survey was conducted to

locate the landmarks reported in various newspaper items and eyewitness accounts published in a UNESCO booklet by

Kakar et al. 2015. The coordinates of these landmarks were used to extract the inundation parameters using Google Earth.

## 3.1 Gwadar

The city of Gwadar is one of the major coastal cities along the coast of Pakistan. The recently built deepwater port has added to the importance of the city. Gwadar is also the hub of Gwadar district today that in itself consists of four sub-districts; Gwadar, Pasni, Ormara and Jiwani.

In 1945 Gwadar district consisted of only Peshkan, Sur, Nigor and Pleri along with Gwadar city (see **Fig. 2** (b)). According to the 1931 census report of India (Vol. I, Part I), chapter 1, page 13, Gwadar had been excluded from the census of India because of being in possession of the Sultan of Muscat. Gwadar was in possession of the Sultanate of Oman from 1734 to 1958. In 1945, the population of Gwadar town was 5875 according to Records of Oman 1867 – 1947 (see **Fig. 2** (a)). For the same reason, no information on the damages was found in Government reports of Baluchistan nor much was reported in Indian newspapers regarding Gwadar. According to a handwritten letter by the Sultan of Oman (Sa'eed Bin Taimoor), Gwadar suffered estimated financial damages of approximately 70,000 rupees and four lives were lost (**Fig. 2** (c)). The letter has previously been translated as "Five nights ago, an earthquake occurred before dawn time, though no damages happened here as the earthquake was subtle, but the sea rose higher than usual to the point that it entered in the wadi that is behind Masjid Al-Khor mosque at the wadi and news have been received about this earthquake from Al-Hind (India) and Makran, and that Gwadar had been greatly affected and the losses have reached approximately 70,000 Rubbiyya and four have been killed, and it is all in the hands of God." by (Hoffmann et al., 2013).

The main source of information at Gwadar is eyewitness accounts (**Table 1**) because of the absence of written history. The eyewitnesses along the coast were interviewed at the beginning of this decade and are compiled and published in the form of a UNESCO booklet by Kakar et al. 2015. These eyewitness accounts form the basis of assessing the approximate runup and inundation extents at Gwadar town. From eyewitness accounts, the places and landmarks that were reported as the inundation extent or being inundated are mapped and shown in **Fig. 3**. Mulla Band and Shadu band, the two dams are the highest landmarks that were identified to be inundated by eyewitness accounts. The maximum runup elevation is found at Jamat Khana (11 m). All the points indicate a runup elevation of 5 to 11 m approximately and inundation extent to be in between 200 to 900 m from the eastern bay (**Fig. 3**) whereas none of the eyewitness accounts reports inundation along the western bay other than Master Abdul Rasheed stating, "Water came from the east and crossed to the other side." The wave was reported to be as high as minaret or to be 3–3.6 m by the eyewitnesses.

APPENDIX "A"
-----------

P O P U L A T I O N.

| | British Subjects | | | Muscat Subjects | |
|---|---|---|---|---|---|
| | Aghakhani Khojas. | Hindus. | Miscellaneous | Arabs | Baluchis. |
| Gwadur Town | 400 | 120 | 305 | 50 | 5,000. |
| Peshkan. | - | - | 20 | 2 | 500. |
| Sur. | - | - | 30 | - | 300- |
| Nigor. | - | - | 15 | - | 1,000. |
| Pleri. | - | - | 1.(Haji Gharib Shah a Baluch Pir or Saint) | - | 30. |
| Total | 400 | 120 | 371 | 52 | 6,830 |

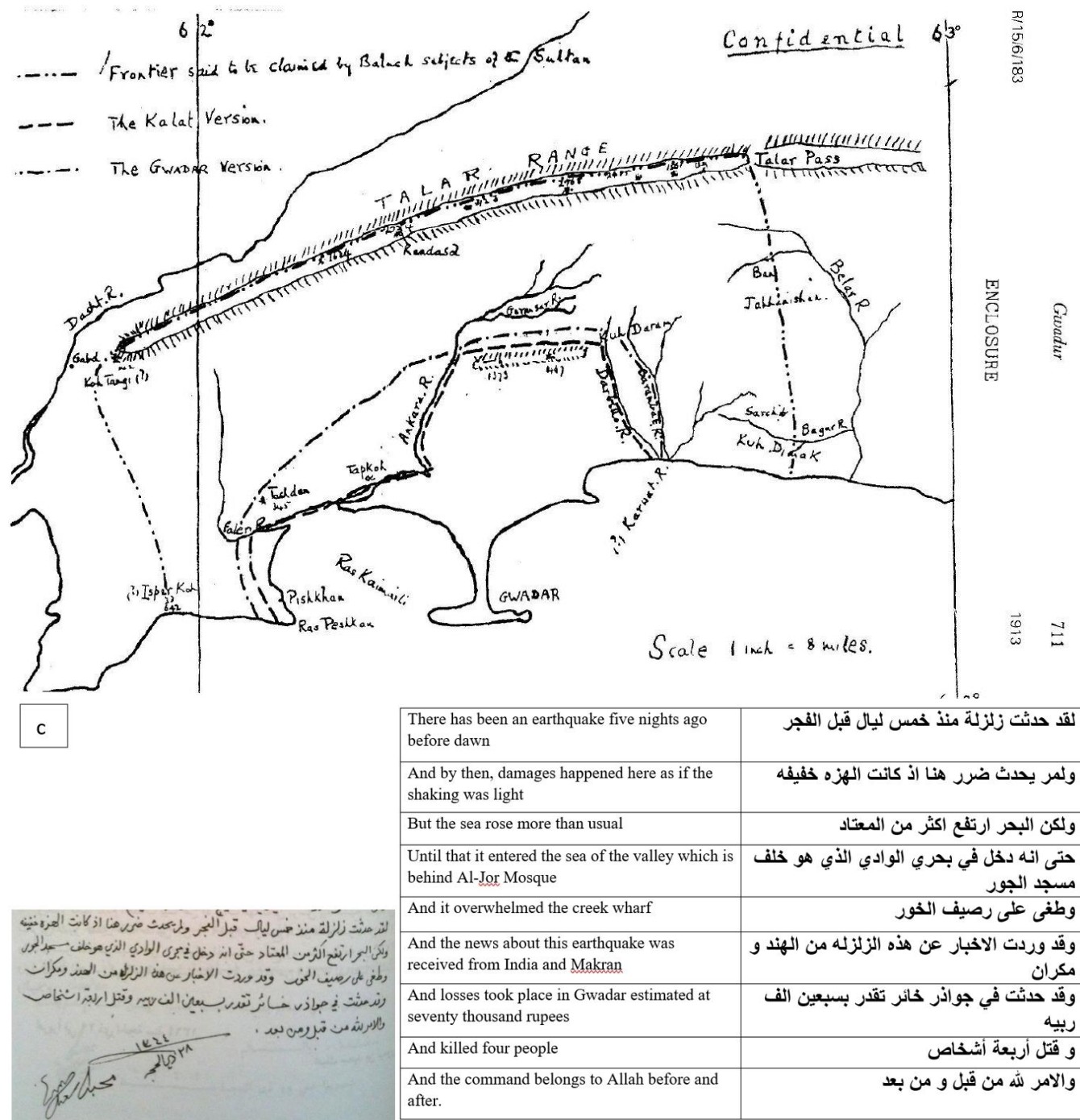

| There has been an earthquake five nights ago before dawn | لقد حدثت زلزلة منذ خمس ليال قبل الفجر |
|---|---|
| And by then, damages happened here as if the shaking was light | ولمر يحدث ضرر هنا اذ كانت الهزه خفيفه |
| But the sea rose more than usual | ولكن البحر ارتفع اكثر من المعتاد |
| Until that it entered the sea of the valley which is behind Al-Jor Mosque | حتى انه دخل في بحري الوادي الذي هو خلف مسجد الجور |
| And it overwhelmed the creek wharf | وطغى على رصيف الخور |
| And the news about this earthquake was received from India and Makran | وقد وردت الاخبار عن هذه الزلزله من الهند و مكران |
| And losses took place in Gwadar estimated at seventy thousand rupees | وقد حدثت في جوادر خائر تقدر بسبعين الف ربيه |
| And killed four people | و قتل أربعة أشخاص |
| And the command belongs to Allah before and after. | والامر لله من قبل و من بعد |

**Fig. 2 (a) Population of Gwadar in 1945 from Records of Oman 1867 – 1947. (b) Old map of Gwadar from a letter written by**
**Lieut. Col. J. Rasmay, agent to the Governor General and Chief Commissioner in Balochistan to mark the boundary of Gwadar**

**and Kalat in 1913, printed in Records of Oman 1867-1947. (c) An excerpt of a letter by Sultan of Oman, Sa'eed bin Taimoor along**
**with transliteration of the excerpt.**

 **Table 1 Summary of eyewitness accounts. Here EQ stands for earthquake. Wave heights are not from some datum but are personal interpretation of the interviewee.**

| Name | Age in 1945 (yrs) | No. of waves | Largest wave | Reported wave heights | Reported arrival times of waves | | Inundation extent/depth |
|---|---|---|---|---|---|---|---|
| **Eyewitnesses at Gwadar** | | | | | | | |
| Amina | 20 | – | – | High as minaret | – | – | Mulla Band, Shadu band, ashkoki, Chanali were completely inundated. Waja Khizer, area infront of Koh e Batil was also inundated. |
| Mulla Murad Mohammad* | – | – | – | 3–3.6 m | – | – | – |
| Hassan Ali* Souhail | – | – | – | – | – | – | Water Jammat Khana (15 feet deep), WAPDA house was inundated and area where Agha khani community lived was also inundated. |
| Master Abdul Majeed | 7-8 | – | – | – | – | – | Water came from east and crossed to the other side. The water also went southward to graveyard near Koh-e-Batil. |
| Hasan Ali* | – | – | – | – | – | – | Water came from east and went towards Mulla Band. Jammat Khana was used as shelter as the building was strong. |
| **Eyewitnesses at Pasni** | | | | | | | |
| Shamsi Mai | 16-17 | – | – | 6–7.6 m | – | – | 2-3 km inland |
| Master Abdul Rasheed | 12 | – | 2nd | – | Before 6:00 am | Around 6:00 am | Few km inland |
| Sakhi Dad | 10-12 | 3 | 3rd | 6–7.6 m | 6:00 am | – | – |
| Qadir Buksh* Kushesh | 5 | – | – | ~ 4.5 m | – | – | – |
| Ajyani Guli | 11 | 3 | | – | – | – | – |
| Khudi Dost | 10-15 | – | – | – | 30 min after EQ | – | Part of Wadsar drowned. |
| Karim Buksh | 13 | 7 or 8 | | – | 6:00 | – | Father's boat was placed by tsunami on the top of mosque. |
| Haroon* | ~1.4 | 3 | | 18, 12, 9 m for 3 waves | – | – | – |
| Rabuk (Rabia) | 5-6 | – | – | – | – | – | Water damaged many houses and a mosque. |
| Ganj Buksh | 14-15 | – | – | – | – | – | destroyed houses, boats, and debris nearly as far inland as Paraag. Many houses and boats were stranded beside Jaddi Hill |
| **Eyewitnesses at Ormara** | | | | | | | |

| Name | | | | | | | Description |
|------|---|---|---|---|---|---|-------------|
| Dildar Sahab | 12 | 3 | _ | _ | _ | _ | Naik Noor Mohammad Dargah inundated with 4 feet deep water. |
| Qadir Buksh | 15-16 | _ | _ | _ | 1-1.5 hrs after EQ | _ | Water went about as far as the present high school and reached the Naik Noor Mohammad Dragah. |
| Madni | 10-11 | _ | _ | 4 m | 30 min after EQ | _ | _ |
| Shamsudin | 6 | _ | _ | _ | 30 min after EQ | _ | _ |
| Master Fateh Mohammad Baloch | 15 | 3 | 3rd | _ | 5 a.m | _ | Water reached Naik Noor Mohammad Dargah. *Gaali*, an Indian cargo boats wreckage was carried to Soorani Stream. |
| Guli | 8 | _ | _ | _ | _ | _ | Water reached Naik Noor Mohammad dargah (knee deep). Family took refuge where now is Teshil Municipal Office. |
| Lari | 11 | _ | 1st | _ | _ | _ | Water reached Naik Noor Mohammad Dargah. Water reached the area where present Fisheries Office is. |
| Sualeh | 12-14 | _ | _ | _ | 30 min after EQ | _ | A lot of big fish like sharks and whales were brought on shore near the Customs House. There were dead bodies where the Fisheries Office is now. |

*learnt about the event through their elders.

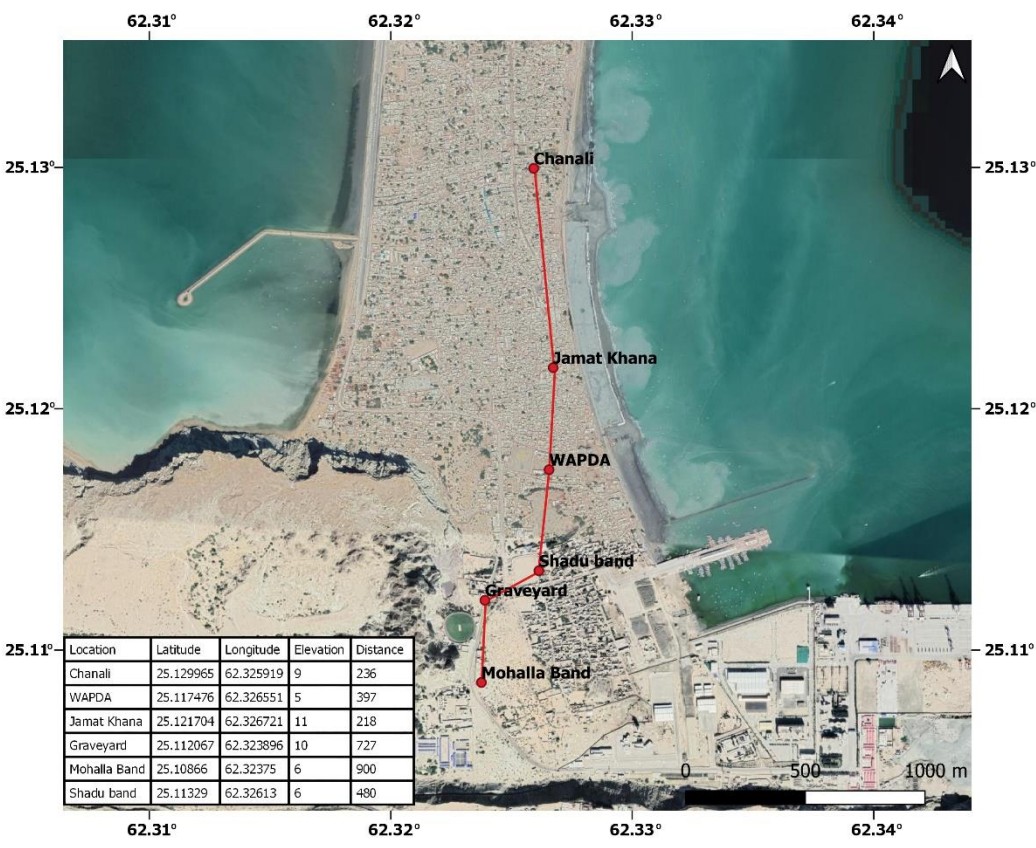

| Location | Latitude | Longitude | Elevation | Distance |
|---|---|---|---|---|
| Chanali | 25.129965 | 62.325919 | 9 | 236 |
| WAPDA | 25.117476 | 62.326551 | 5 | 397 |
| Jamat Khana | 25.121704 | 62.326721 | 11 | 218 |
| Graveyard | 25.112067 | 62.323896 | 10 | 727 |
| Mohalla Band | 25.10866 | 62.32375 | 6 | 900 |
| Shadu band | 25.11329 | 62.32613 | 6 | 480 |

**Fig. 3 Locations as identified by eyewitness accounts to have been inundated by the 1945 tsunami, plotted on © Google Satellite image. The line shows a crude estimate of inundation extents.**

**3.2 Pasni**

The City of Pasni still remains small even today. It lies on the Makran Coast of the Arabian Sea about 450 km from Karachi. Administratively, Pasni is the headquarter of the Pasni sub-division of Gwadar district that includes Pasni and Ormara Tehsils (tehsil - county) as well as Astola Island which lies 40 km ESE of Pasni, in the Arabian Sea. According to the census of India, Volume IV, Baluchistan (pp. 12) in 1931 total population of Pasni was 1989 (Male: 1090 and Female: 899) which grew to3616 (Male: 1852 and Female: 1764) in 1941 (Census of India, Volume XIV, Baluchistan, pp. 14). Therefore, it is estimated that the population of Pasni in 1945 would have been in the 4000s.

The Baluchistan Agency Administration Report 1945-46 in many of its sections described the devastation caused by a tidal wave that was preceded by an earthquake. Part I of Baluchistan Agency Administration Report 1945-46, reports of a severe

earthquake on the coast of Makran and Lasbela on 28th November 1945 at 3:30 am. It further reports that Ormara and Pasni suffered substantial damages. According to the report around 7:00 am, 30 feet high tidal wave struck Pasni, submerging the entire town while claiming 47 lives (Fig. 4).

A very severe earthquake occurred on the coast of Mekran and Lasbela State on the 28th November, 1945. The shock began at 3-30 A.M. At Ormara considerable damage was caused to buildings and 71 lives were lost. At Pasni a tidal wave 30 feet high arose at 7-0 A.M. and submerged the whole town. 47 lives were lost. Both at Pasni and Ormara a large proportion of fishing craft and tackle was destroyed.

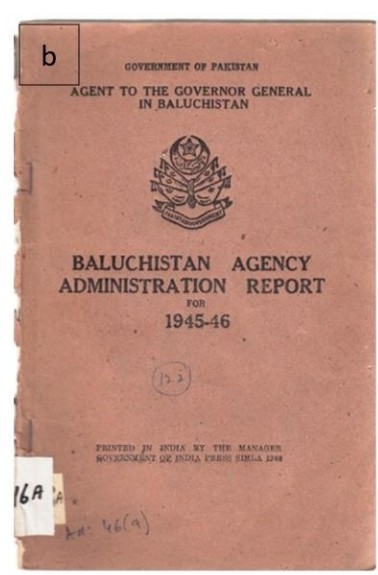

5. *The Pasni Earthquake.*—On the night of 28/29th November 1945 a serious earthquake occurred in the Sea off the South-Western coast of Mekran which was closely followed up by High tide of water that completely destroyed and washed off the once prosperous and industrious town of Pasni. The village of Kalmat was also seriously damaged. The total casualties to human-beings were 46 dead and several injured, while the loss of property amounted to Rs. 13,33,000.

An appeal for funds to afford relief to the sufferers was made to all his subjects, by His Highness the Khan, as a result of which a sum of Rs. 40,000 was collected in the State, which together with the generous donation of Rs. 60,000 from the Baluchistan Administration, was distributed among those of the victims of the tragedy who were found to be really in need of help.

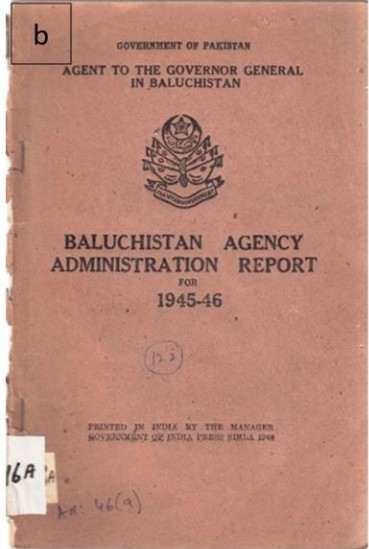

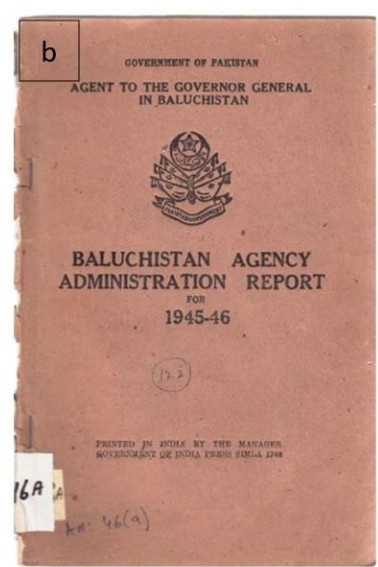

5. *The Pasni Earthquake.*—On the night of 28/29th November 1945 a serious earthquake occurred in the Sea off the South-Western coast of Mekran which was closely followed up by High tide of water that completely destroyed and washed off the once prosperous and industrious town of Pasni. The village of Kalmat was also seriously damaged. The total casualties to human-beings were 46 dead and several injured, while the loss of property amounted to Rs. 13,33,000.

An appeal for funds to afford relief to the sufferers was made to all his subjects, by His Highness the Khan, as a result of which a sum of Rs. 40,000 was collected in the State, which together with the generous donation of Rs. 60,000 from the Baluchistan Administration, was distributed among those of the victims of the tragedy who were found to be really in need of help.

**Fig. 4 (a) Excerpts of Baluchistan Agency Administration Report, 1945 – 1946, Part I. (b) Excerpts of Baluchistan Agency**
**Administration Report, 1945 – 1946, Appendix XI, pp. 59 and 60.**

Appendix XI Kalat State, of the same, reports, "A serious earthquake occurred in the Sea off the South-Western coast of
Makran which was closely followed up by a High Tide of water that completely destroyed and washed off the once
prosperous and industrious town of Pasni." The financial damages and relief efforts at Pasni are also mentioned. It further
states that the Khan of Kalat made an appeal for funds to provide relief to the sufferers which resulted in a substantial
amount that was afterwards distributed among the people at Pasni (**Fig. 5**).
This unfortunate event was widely reported by many newspapers around the world but it was most extensively covered by
"Times of India." Times of India on Friday, 30th November 1945 reported seawater rushed into the town of Pasni and
washed away a good number of people. Government buildings including Post and Telegraph office and rest house were
washed away. Times of India on Saturday, 1st December 1945 reported, "the town of Pasni is a vast sheet of water with only
housetops being visible….Custom House is reported to have been damaged". Times of India on 6th December 1945 reported
that Mr. J. L. Jerath, Director Posts and Telegraphs, Sind and Baluchistan, who had been on H.I.M.S. Hindustan, a naval ship
sent to Pasni and Ormara for relief work, upon his return from Pasni and Ormara said that 80% of the huts at Pasni and 60%
of the huts at Ormara are estimated to be washed away by the tidal wave (**Fig. 5**). Sind Observer on 6th December 1945
reported for Pasni, "The whole village has been totally razed to the ground…..Customs goods and other properties including
furniture were carried away by the tidal wave to the other extreme of the village. About 7,000 people here are homeless."

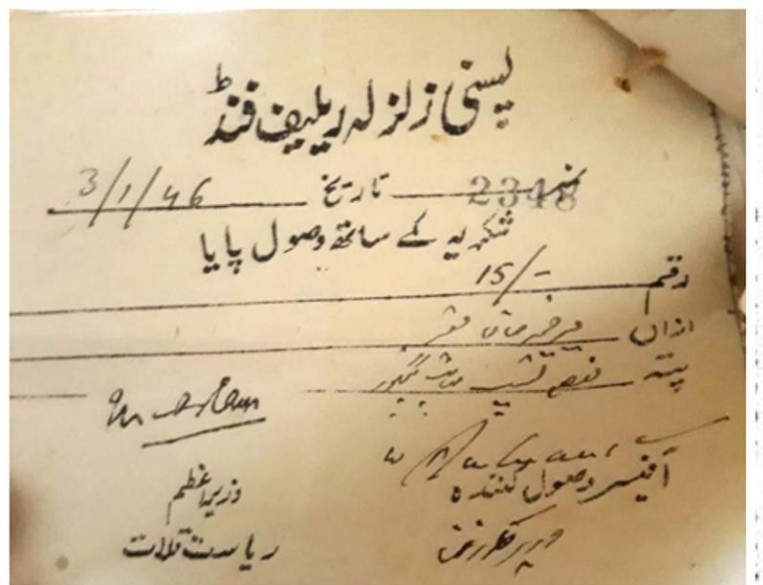

Mr. Jerath, who left Karachi on Saturday by H.M.I.S. HINDUSTAN, said that at Ormara, most of the huts had been washed away by the tidal wave. At Pasni the destruction was even more widespread. It is estimated that about 60 per cent of the huts have been washed away at Ormara and 80 per cent at Pasni. About 100 persons have probably been killed at either place. There were a number of injured and doctors on board the H. M. I. S. KARACHI, which was sent on Thursday, and H. M. I. S. HINDUSTAN, which left Karachi on Saturday, treated them. Five serious cases, needing hospital treatment, have been brought to Karachi by the HINDUSTAN.

150

**Fig. 5 Relief efforts at Pasni and Ormara. Slip for an amount of 15 PKR of Pasni Relief Fund received by a survivor of 1945 tsunami (on the right). Times of India clipping showing that Director Post and Telegraph went on the H.M.I.S. Hindustan to Pasni and Ormara (on the left).**

The inundation extents and runups were not reported in any of the government reports and newspaper items. The places, Rest House and Post and Telegraph office reported by Times of India as being washed off by the tsunami; were located through an old map of the Pasni city, from 1943 (a quarter-inch sheet of by the Survey of India. G41-P Turbat, interim edition 1941, reprinted April 1943, scale 1:253,440), (**Fig. 7**). PTO was found to be approximately 460 m and Rest House at 570 m from the shoreline at that time. The shoreline of Pasni has changed since 1945, not only as a result of erosion and deposition of sediments but also because of the event itself as it is reported by many eyewitnesses that part of Pasni slid underwater.

The extents of inundation based on field survey following the eyewitness accounts and reported landmarks therein are approximately 300 to 700 m from the shoreline whereas the runup elevations are between 4 – 14 m (**Fig. 6**). Among these points, Wadsar is the one closest to shore and also has minimum runup elevations but as this area was reported by several eyewitnesses to have been drowned or slid underwater because of the event therefore we expect that location of Wadsar is not the actual inundation extent but it is rather an area which was inundated (see **Fig. 6**). Moreover, the number of waves as per the eyewitness accounts were three.

168

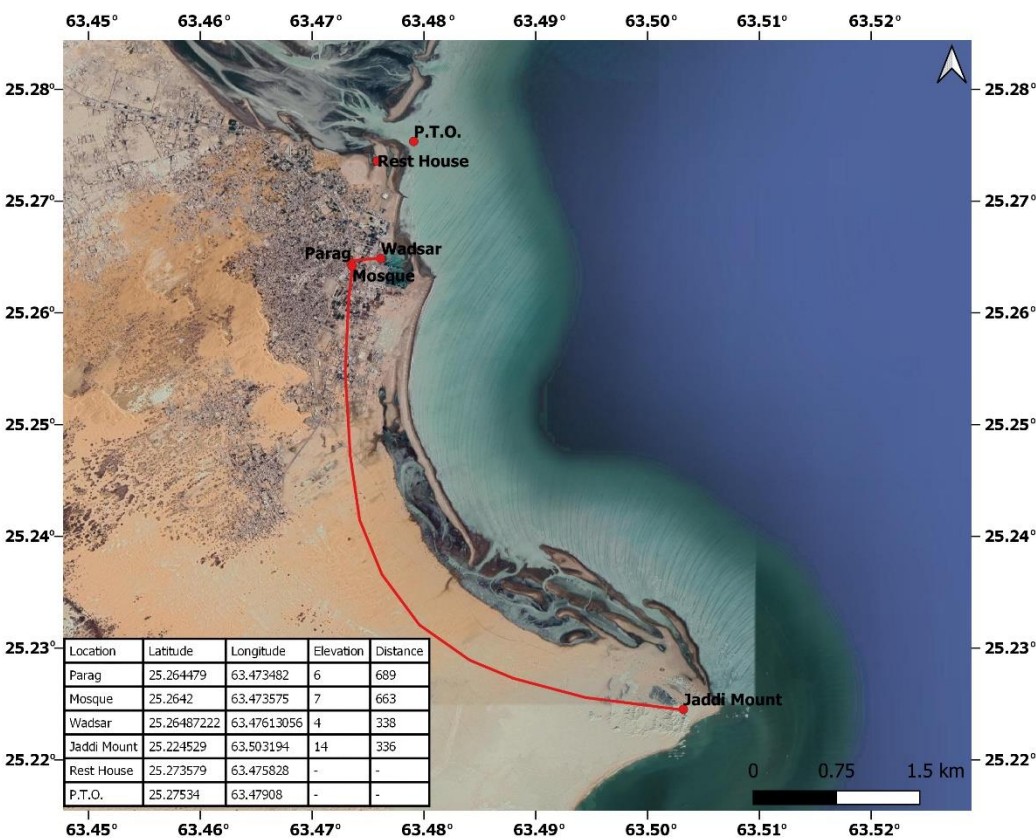

| Location | Latitude | Longitude | Elevation | Distance |
|---|---|---|---|---|
| Parag | 25.264479 | 63.473482 | 6 | 689 |
| Mosque | 25.2642 | 63.473575 | 7 | 663 |
| Wadsar | 25.26487222 | 63.47613056 | 4 | 338 |
| Jaddi Mount | 25.224529 | 63.503194 | 14 | 336 |
| Rest House | 25.273579 | 63.475828 | - | - |
| P.T.O. | 25.27534 | 63.47908 | - | - |

169

Fig. 6 Locations as identified by eyewitness accounts to have been inundated by the 1945 tsunami, plotted on © Google Satellite image. The line shows a crude estimate of inundation extents. The points which have not been joined through the line were identified from newspaper accounts.

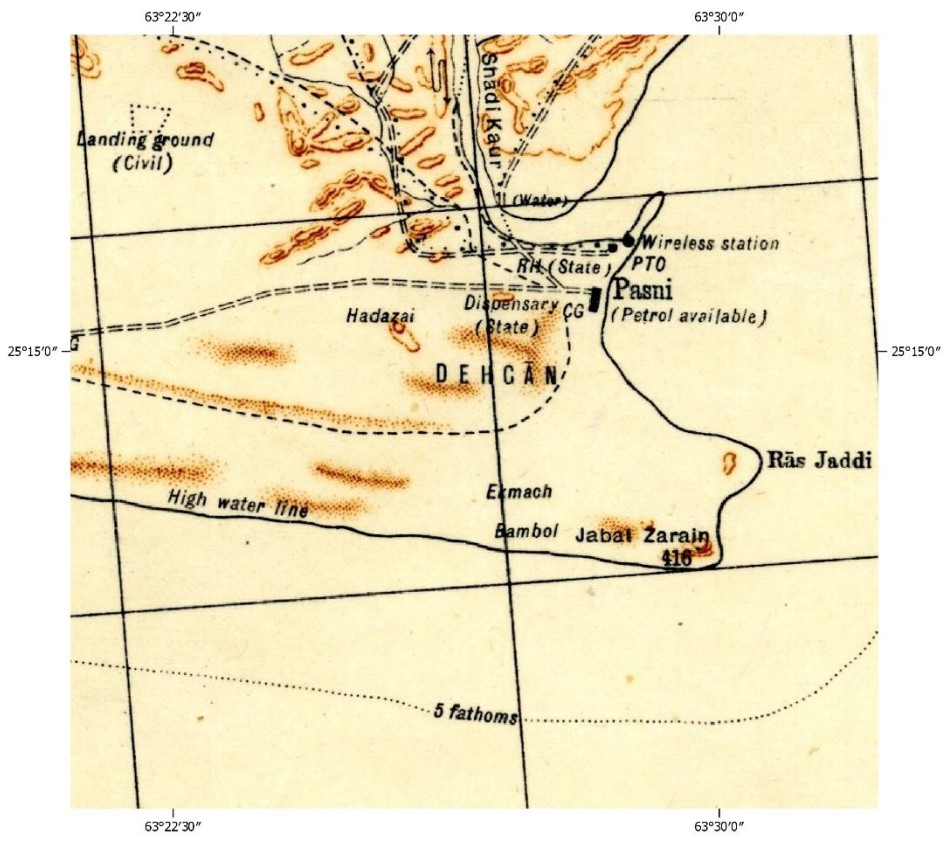

173

**Fig. 7 Old map of Pasni. An excerpt from a quarter-inch sheet by the Survey of India. G41-P Turbat, interim edition 1941, reprinted April 1943, scale 1:253,440.**

176

### 3.3 Ormara

Ormara still is not very populous but it is an important city of Gwadar district along the Makran coast. Ormara in 1945 came under the Las Bela state and was part of British Balochistan. The first year for which the population for the city of Ormara could be found during the study is 1981. According to a report of Pakistan bureau of statistics in 1981 total population of Ormara was 8265. Therefore, it can be speculated that the city of Ormara had a population of only 1,000s in 1945.

In the Baluchistan Agency Administrative Report Appendix XII, the damages by the 1945 event are reported stating that it resulted in 78 deaths and 165 people were injured though it is unclear whether the tsunami caused the fatalities or the earthquake itself caused the deaths (**Fig. 8**).

Devastation at Ormara was not much less than the devastation at Pasni. As reported in Times of India, 6[th] December, Mr. Jerath, Director Posts and Telegraph estimated 60% of huts to have been washed away by tsunami at Ormara. Dawn reported on 2[nd] December 1945 that the town of Pasni was completely flat and the condition at Ormara is no different from Pasni.

### APPENDIX XII.
## ADMINISTRATION REPORT OF LAS BELA STATE FOR THE YEAR
### 1945-46.
#### CHAPTER I.—*General and Political.*

4. A severe earthquake occurred at Ormara on the 27th November 1945 resulting in 78 deaths and injuries to 165 persons. In addition, 12 persons were found missing. The loss of property is estimated to range between three to four lakhs of rupees. Relief measures were taken at the time.

**Fig. 8 Excerpts of Baluchistan Agency Administration Report, 1945 – 1946, Appendix XII**

Eyewitnesses remembered the arrival of three waves after the earthquake and destruction of an Indian cargo boat, *Gaali* and the wreckage being carried to Sorani stream. The waves arrived either an hour or an hour and a half after the earthquake. The accounts have been quantified to get inundation extent and runup at Ormara, through a ground survey. It is found that the maximum runup elevation is approximately 11 m and the maximum inundation extent is almost 2.5 km (**Fig. 9**).

The Post and Telegraph Office (PTO) was reported by the Times of India to have been inundated during the 1945 event. The PTO was located through an old map of the city (a quarter-inch sheet by the Survey of India. G41-Q Ormara, second edition 1937, scale 1:253,440) and was found to be approximately 1 km from the shoreline.

Interviews of local fishermen at Ormara in the 1970s, reported in (Page et al., 1979) provided evidence of uplift at Ormara due to the 1945 earthquake which is interpreted by the author to be around 2 m. The same is evident by the interview of Qadir Buksh, "The shoreline shifted. Before the event the shore was inland of where it is today." (Kakar et al., 2015a).

201

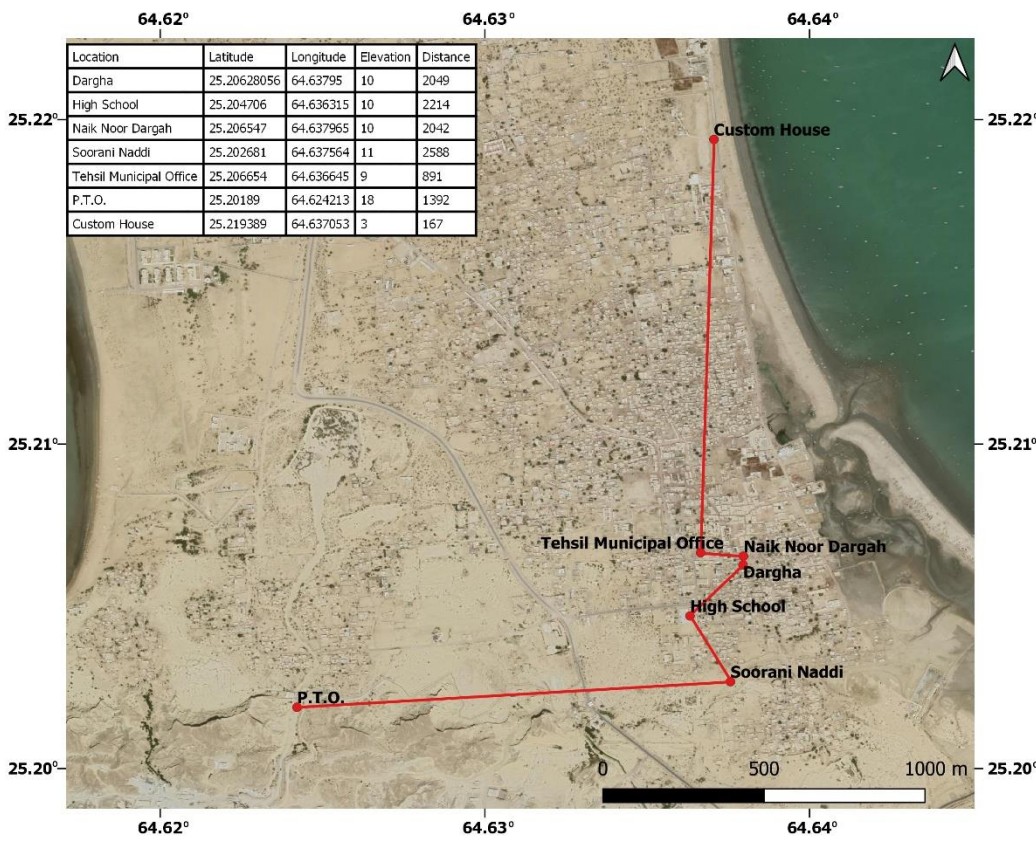

| Location | Latitude | Longitude | Elevation | Distance |
|---|---|---|---|---|
| Dargha | 25.20628056 | 64.63795 | 10 | 2049 |
| High School | 25.204706 | 64.636315 | 10 | 2214 |
| Naik Noor Dargah | 25.206547 | 64.637965 | 10 | 2042 |
| Soorani Naddi | 25.202681 | 64.637564 | 11 | 2588 |
| Tehsil Municipal Office | 25.206654 | 64.636645 | 9 | 891 |
| P.T.O. | 25.20189 | 64.624213 | 18 | 1392 |
| Custom House | 25.219389 | 64.637053 | 3 | 167 |

202

**Fig. 9 Locations as identified by eyewitness accounts to have been inundated by the 1945 tsunami, plotted on © Google Satellite image. The line shows a crude estimate of inundation extents. The point which has not been joined through the line was identified from newspaper accounts.**

## 4. Results and Discussion

The historical accounts for large earthquakes along the Makran Subduction zone are sparse and disputable. Nevertheless, the possibility of large earthquakes cannot be ruled out. With Megacities such as Karachi (Pakistan) and Mumbai (India) and many other growing coastal cities such as Gwadar (Pakistan), Chabahar (Iran) and Batinah (Oman), the seismic hazard from Makran Subduction Zone and risk of ensuing tsunamis cannot be overlooked. The growing population and large investments in infrastructure along the coasts bordering the Arabian Sea demand reliable risk assessment for tsunami in the region but not enough data is available for the same.

In many cases, historical accounts are a valuable source of information for the reconstruction of past tsunami events (Atwater et al., 2013; Dominey-Howes et al., 2006) where scientific data is not present. We first summarize the description of the 1945 event in newspaper items, historical reports and eyewitness accounts and then use eyewitness accounts and newspaper items combined with a field survey to extract the runups and inundation extents for coastal cities of Pakistan through the reported tsunami observations there-in.

At Gwadar, although there was not much damage the maximum runup is found to be 11 m and the maximum inundation extent is around 900 m. These extents have been derived from the landmarks identified by the eyewitnesses but one of the eyewitnesses (Master Abdul Majeed) also reported, "Water came from the east and crossed to the other side" which is indicative of tsunami engulfing the entire landmass along the east to west stretch. None of the other eyewitnesses reported such inundation, The study does not use this account to conclude that the water might have swept across the entire tombolo as many other survivors had reported water reaching up to certain landmarks only. Another survivor of the event, Amina reported that the "huge wave" did not enter the city. She further reported the water reached the mosque; water was everywhere with no place to go but the water went further than the mosque. She also named some places that were inundated by the tsunami, such as the Mulla band and Shadu band (Kakar et al., 2015b).  The water reaching the Mulla Band, reported by Amina and Hasan Ali might be that they were reporting "Mohalla Band" rather than "Mulla Band" or "Mohalla Band" is the new name of the neighbourhood just beside the Gwadar Miniport which was previously called as "Mulla Band", an area that is very likely to be inundated during the 1945 event. Shadu Band is another neighbourhood beside the new football stadium of Gwadar. In order to be sure if the interpretation of the locations was right, interviewers of the Amina were interviewed as Amina had passed away.

The maximum runup and inundation extent at Pasni as measured are approximately 14 m and 700 m, respectively. The inundation extents are not the actual extents for every point marked on **Fig. 3** but in some cases mark the landmarks that were identified as inundated. Moreover, the shoreline at Pasni has changed drastically since 1945 and the inundation extents for most of the points have been extracted using the recent imagery from Google Earth. Therefore, these two factors can contribute to the fact that the actual inundation extent in 1945 could have been greater than reported here.

At Ormara the maximum runup and inundation extents are approximately 11 m and 2.5 km (from Western Bay after the epicenter from (Byrne et al., 1992) (see **Table 2**). The inundation extent at Ormara is the greatest among all the towns considered in the study although Pasni was much closer to the epicentre. This might be contributed by the fact that Pasni had sand dunes near the town which according to many eyewitnesses saved their lives as it was a place of refuge whereas at Ormara no such natural defence was present beside the town.

**Table 2 Impact of 1945 Makran tsunami along the coastal cities of Pakistan.**

| City | Maximum runup (m) | Maximum Inundation extent (m) | Number of Waves | Maximum Wave Height (m) | Casualties | Financial Damages (Rs.) | Present day equivalent (US $) |
|------|------|------|------|------|------|------|------|

| | | | | | | | |
|---|---|---|---|---|---|---|---|
| Gwadar | 56 | 700 | – | 3–6 | 3–4 | 70,000 | ~453 |
| Pasni | 7.6 | 1000 | 3 | 9.1 | 47 | 1,333,000 | ~8630 |
| Ormara | 11 | 2500* | 3 | – | 76 | 300,000-400,000 | ~1945–2589 |

*from Western Bay

If we take the same population as 1941 (1939) and find the percentage of people who lost their lives to the 1945 tsunami event at Pasni, it is found to be approximately 1.3% (considering the population of 1941 as the nearest estimate of population in 1945). The town of Ormara had an estimated population of nearly 1000 and sustained 76 casualties that give approximately 8% of the population wiped off by the event.

## 5 Conclusions

This paper draws on the eyewitness accounts and newspaper items to estimate the runup and inundation extent at Gwadar, Pasni, Ormara and Karachi. Pasni and Omara were the most severely affected cities. The inundation extent at Ormara is the greatest among all the cities considered in the study although Pasni was much closer to the epicentre. The uncertainty is inherent to the parameters derived here due to reasons such as personal interpretation of the event survivors and survey being conducted after 70 years of the event. Therefore, the inundation parameters presented here may be a crude approximation of the actual parameters but it still paints a picture of the wreck-havoc caused by the 1945 Makran tsunami.

The data collected in the form of eyewitness accounts, archival reports and newspaper accounts from countries bordering the Arabian Sea should be used to draw reliable limits on the source of the earthquake and ensuing tsunami. Similar studies in the neighbouring countries can further facilitate the cause and contribute to reliable risk assessment of the coasts along the Arabian Sea.

The time of arrival of waves at Pasni as reported by multiple survivors was around 6 a.m. whereas only Khudi Dost reports the waves to have arrived almost half an hour after the earthquake (**Table 1**). It is reported in Baluchistan Agency Administration Report (**Fig. 4**), *"At Pasni a tidal wave 30 feet high arose at 7-0 A.M. and submerged the whole town."* Therefore, it is evident that there is a time difference of 2–3 hours between the earthquake and the arrival of the largest wave. This finding is in concordance with the eyewitness accounts from Iran and the finding is reported in (Okal et al., 2015) and with the observation of (Beer and Stagg, 1946). This time delay in the arrival of tsunami is suggestive of some secondary mechanism such as landslide, associated with the earthquake. This can also be the reason why most of the witnesses reported that the 2nd or the 3rd wave as being the highest of the waves that attacked the coast.

The majority of the eyewitnesses along the Makran coast of Pakistan had reported the time of arrival of the tsunami as half an hour after the earthquake. (Beer and Stagg, 1946) reported, "The first tidal observation was made at 9 hr. 47 min. local time, but it was then noted that the tidal-levels were well above their normal value, suggesting that an earlier wave may

indeed have arrived by that time." Therefore, the time reported here by the eyewitnesses as thirty minutes after the earthquake might be the time of arrival of the first wave associated with the earthquake whereas the larger wave generated by an ancillary phenomenon arrived 2–3 hours after the earthquake.

The total number of estimated fatalities associated with the Makran earthquake and ensuing tsunami vary between 300 (Ambraseys and Melville, 1982) to 4000 (https://www.ngdc.noaa.gov/hazards/tsu_db.shtml). The more widely reported number of fatalities is 4000 (e.g., Heck, 1947; Heidarzadeh et al., 2008; Rajendran et al., 2008) but this figure is associated with only the region of Karachi and Indus Delta rather than the Makran coast of Pakistan. According to Times of India, 5th December 1945, the reports of 4000 casualties came from a party of nine congressmen. It was reported only for the 100 miles coast from Karachi to Keti-bunder (a region in Indus Delta). These reports, according to an express letter written by the Chief Secretary to the Government of Sind, to the Secretary to the Government of India were "greatly exaggerated."

Moreover, according to the comment of the Chief Secretary to the Government of Sind on estimates of the loss of lives by congressmen, published in Times of India, 6th December 1945, "They were highly exaggerated. The coastline is sparsely populated. The sub-divisional officials have asked for only small grants for relief, indicating that the damage caused is not as heavy as reported."

## Author Contribution

Hira Ashfaq Lodhi prepared the manuscript, identified the landmarks from eyewitness accounts and newspaper items for the field survey. Shoaib Ahmed conducted the field survey. Haider Hasan searched for archival documents.

## Competing Interests

The authors declare that they have no conflict of interest.

## Funding

The field survey was funded under UNDP project "Tsunami and Earthquake Preparedness in Coastal Areas of Pakistan."

## Acknowledgments

We acknowledge the support extended by Dr. Brian F. Atwater. His critical reviews have improved the manuscript. We also acknowledge the support extended by Dr. Gösta Hoffman for sharing the excerpt of the Sultan's letter. We also acknowledge our reviewers for their input.

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
