# Peer review of "Tsunami heights and limits in 1945 along the Makran coast estimated from testimony gathered seven decades later in Gwadar, Pasni and Ormara"

_Natural Hazards and Earth System Sciences, 2021_

## Referee Comment (RC1)

**Review of "Tsunami heights and limits in 1945..." by Lodhi *et al*.**

*Review by E.A. Okal* (I waive anonymity)

The paper attempts the compilation of a dataset of runup and inundation of the 1945 tsunami at three population centers along the coast of Pakistan, based on the interviews conducted earlier and compiled in the UNESCO report [*Kakar et al.,* 2015].

As such, the paper is a valuable extension of that work, which had reported the interviews in excruciating detail, but had fallen short of transforming them into a scientifically usable database. In this respect, the paper would deserve publication.

→ Unfortunately, it stops short of this goal because it does not provide coordinates (latitude; longitude) for the seven locales at which quantitative estimates are given in Table 1. **The paper cannot be published without this information**, especially since *"The lat/long* [sic, should use the word "coordinates"] *of these landmarks were used to extract the inundation parameters using Google Earth"* [Page 3, Line 64].

Additional shortcomings of the paper are listed below.

*Emphasize major conclusions*

**1.** A major conclusion of this study seems to be that the number of fatalities was at most 150 (Table 2). This is in contrast to the figure of 4000 reported by the NOAA Tsunami Database. The discussion in the present paper would suggest populations of about 6000 in Gwadar (Line 72), 4000 in Pasni (Line 104, even though a newspaper reports 7000 people homeless) and perhaps 1000 in Ormora (Line 160), for a total of 11,000. The rest of the coast was probably very scarcely populated. A death toll of 4000 would amount to 1/3 of the total population, and would be an extremely high rate with long-lasting consequences on the economy of the province. It would probably have been mentioned repeatedly during the interviews of the (then very few) survivors. In this context, the NOAA figure is most probably grossly overestimated.

\* Some discussion of this finding should be provided in the paper.

**2.** The newspaper clipping on Figure 3 contains an extremely important datum, namely that the tsunami reached Pasni around 07:00. The earthquake is known to have taken place at 21:57 GMT (on 27-NOV-1945), which agrees with the felt report at 03:30 (28-NOV) given IST (in use in 1945) = GMT + 5:30. There is therefore a delay of about three hours in the arrival of the tsunami.This is in line with the delay of ~2.5 hours reported by witnesses on the Iranian side [*Okal et al.* 2015], and also with the famous observation of the tsunami in the Seychelles [*Beer and Stagg,* 1946]. *This provides one more piece of evidence that the tsunami* (or at least its main component) *was generated by an ancillary phenomenon, most probably a landslide triggered by the earthquake, but with a significant time gap.*

Arguably, the report on Line 171 suggests a shorter time gap, but it has been our experience that the perception of time by witnesses oftens lacks precision. The fundamental point here is that the earthquake was felt in the middle of the night and the tsunami arrived by daylight.

→      At any rate, this point should be discussed in the paper.

**Insufficient referencing**

**3.**      The authors fail to mention the quantitative compilation carried out across the border in Iran by *Okal et al.* [2015].

**4.**      The authors mention *Atwater et al.* [2013] as a reference to tsunami surveys conducted for historical tsunamis many years after the event. However, this technique was pioneered a decade earlier for the 1946 Aleutian tsunami by *Okal et al.* [2002], which should probably be referenced.

**5.**      The authors fail to reference the authoritative work of *Ambraseys and Melville* [1982] from which most of the information in *Dominey-Howes et al.* [2006] and *Pararas-Carayannis* [2006] is derived.

**6.**      *Page 2, Line 54*

The reference to *Byrne and Davis* [1992] should not include first names (by the way, Dr. Byrne's is misspelt), and should really be *Byrne et al.* [1992] since the full authorship of that paper includes Professor L.R. Sykes, whose name has been reduced to his initials (L.R.S.) in the reference list.

**Major problems: Tables; Figures**

**7.**      The coordinate scales on Figures 2, 5, 8 are completely out of range. Note that the longitude scales from 26°E through 176°E to 34°W. The latitudes are similarly extravagant.

\*      The captions for these figures should name the specific cities.

**8.**      There are some obvious discrepancies in the ages quoted for the witnesses. Notwithstanding the difficulty of obtaining their ages (as discussed, *e.g., Okal et al.* [2015], the latter should be consistent.

Note for example the case of Ms. Amina on Table 1. She is quoted as being ≥ 100 yrs. old at the time of the interview

         (Note that 100+ is not a proper scientific notation. Use the symbols >, ≥, etc.)

but only 20 in 1945. She would then have been born in 1925, which would make her at most 90 in 2015 or 95 in 2020.

Similarly, Ajyani Guli cannot have been 11 in 1945 (b. 1934) and already ≥90 at the time of the interview.

**Other issues**

**9.**      All information should be metric. Convert feet to meters throughout.

**10.**      *Page 2, Line 32*

The authors should emphasize the difference between the 2013 event for which a definitive tsunami requiring a landslide was observed, and the landslide on the Owen Ridge [*Rodriguez et al.,* 2013] which is well documented, but for which the tsunami attacking

Oman can only be inferred.

**11.** *Page 2, Line 57*

The earthquake was followed by five **recorded** aftershocks. There probably were many more.

**12.** *Section 3.1*

There are references to Table 0.1) and Fig 0.2. This needs to be corrected.

**13.** *Figure 1*

Part (a) of the figure is hardly legible. I had to iuse a magnifying glass to decipher it.

Translate the material in Arabic (or is it another language?) in Part (c), which will otherwise be completely useless to most of the readership.

**14.** *Page 14, Table 2, Last Column*

The figure 13,33,000 makes no sense (even though it seems to be quoted directly from the Baluchistan Agency Adminstration Report on Figure 3). Does this mean 1,333,000 or 13,330,000 ? At any rate, if a proper rendition of this number is given, then an exchange rate to a more universal currency should be included (*e.g.,* Rp. XXXXX, equivalent to present-day YYYYY £ or US $ ZZZZ or TTTT €).

**15.** The English of the paper should be improved throughout. There are articles, occasionally verbs, missing. Dr. Brian Atwater's name is misspelt in the Acknowledgments, etc.

**Additional references cited**

Ambraseys, N.N., and C.P. Melville, *A history of Persian earthquakes,* Cambridge Univ. Press, 219 p., 1982

Beer, A., and J.M. Stagg, Seismic sea-wave of November 27, 1945, *Nature,* **158,** 63, 1946.

Okal, E.A., C.E. Synolakis, G.J. Fryer, P. Heinrich, J.C. Borrero, C. Ruscher, D. Arcas, G. Guille, and D. Rousseau, A field survey of the 1946 Aleutian tsunami in the far field, *Seismol. Res. Letts.,* **73,** 490–503, 2002.

Okal, E.A., H.M. Fritz, M.A. Hamzeh, and J. Ghasemzadeh, Field survey of the 1945 Makran and 2004 Indian Ocean tsunamis in Baluchistan, Iran, *Pure Appl. Geophys.,* **172,** 3343–3356, 2015.

---

## Author Comment (AC4)

**Response to reviews of RC1**

We are very thankful to the Dr. Emile Okal for his constructive reviews that have improved the paper. Appended below, in italics and indented, are the full comments from RC1. Our responses follow each of the comments and new text intended to be added to manuscript is in bold.

> *A major conclusion of this study seems to be that the number of fatalities was at most 150 (Table 2). This is in contrast to the figure of 4000 reported by the NOAA Tsunami Database.The discussion in the present paper would suggest populations of about 6000 in Gwadar (Line 72), 4000 in Pasni (Line 104, even though a newspaper reports 7000 people homeless) and perhaps 1000 in Ormora (Line 160), for a total of 11,000. The rest of the coast was probably very scarcely populated. A death toll of 4000 would amount to 1/3 of the total population, and would be an extremely high rate with long-lasting consequences on the economy of the province. It would probably have been mentioned repeatedly during the interviews of the (then very few) survivors. In this context, the NOAA figure is most probably grossly overestimated.*
> *\* Some discussion of this finding should be provided in the paper.*

[revised manuscript text omitted]

*The authors fail to mention the quantitative compilation carried out across the border in Iran by Okal et al. [2015].*

Point taken. This will be included in the revised version, in both introduction and conclusions sections.

*The authors mention Atwater et al. [2013] as a reference to tsunami surveys conducted for historical tsunamis many years after the event. However, this technique was pioneered a*

*decade earlier for the 1946 Aleutian tsunami by Okal et al. [2002], which should probably be referenced.*

Point taken. This will be included in the revised version.

*The authors fail to reference the authoritative work of Ambraseys and Melville [1982] from which most of the information in Dominey-Howes et al. [2006] and Pararas-Carayannis [2006] is derived.*

This will be included in the revised version.

*Page 2, Line 54*
*The reference to Byrne and Davis [1992] should not include first names (by the way, Dr. Byrne's is misspelt), and should really be Byrne et al. [1992] since the full authorship of that paper includes Professor L.R. Sykes, whose name has been reduced to his initials (L.R.S.) in the reference list.*

Corrected on Lines 54 and 57 also on line 210. The reference list has also been corrected.

*The coordinate scales on Figures 2, 5, 8 are completely out of range. Note that the longitude scales from 26°E through 176°E to 34°W. The latitudes are similarly extravagant.*
*\* The captions for these figures should name the specific cities.*

Unfortunately, we missed on noting the error in the coordinates range. Thank you very much for pointing out. We will correct it in the revised version.

*There are some obvious discrepancies in the ages quoted for the witnesses. Not withstanding the difficulty of obtaining their ages (as discussed, e.g., Okal et al. [2015], the latter should be consistent.*
*Note for example the case of Ms. Amina on Table 1. She is quoted as being $\geq 100$ yrs. old at the time of the interview*
*(Note that 100+ is not a proper scientific notation. Use the symbols >, $\geq$, etc.)*
*but only 20 in 1945. She would then have been born in 1925, which would make her at most 90 in 2015 or 95 in 2020.*
*Similarly, Ajyani Guli cannot have been 11 in 1945 (b. 1934) and already $\geq 90$ at the time of the interview.*

*All information should be metric. Convert feet to meters throughout.*

Agreed that there are discrepancies in age. The ages in 1945 and at the time of interview, were quoted directly from "Remembering the 1945 Makran Tsunami; interviews with survivors beside the Arabian Sea". We should have been more skeptical towards the ages and should have discussed the discrepancies and the reasons for it in the paper. This will be improved in the revised manuscript.

The information should be in the same unit system, thank you for pointing out. We will do the conversion.

> *Page 2, Line 32*
> *The authors should emphasize the difference between the 2013 event for which a definitive tsunami requiring a landslide was observed, and the landslide on the Owen Ridge [Rodriguez et al., 2013] which is well documented, but for which the tsunami attacking Oman can only be inferred.*

Change made.

> *Page 2, Line 57*
> *The earthquake was followed by five **recorded** aftershocks. There probably were many more.*

Point taken and the word "recorded" added to the sentence on Page 2, Line 57.

> *Section 3.1*
> *There are references to Table 0.1) and Fig 0.2. This needs to be corrected.*

Agreed and will be corrected.

> *Figure 1*
> *Part (a) of the figure is hardly legible. I had to use a magnifying glass to decipher it. Translate the material in Arabic (or is it another language?) in Part (c), which will otherwise be completely useless to most of the readership.*

Agreed. Figures will be replaced to make these more meaningful.

> *Page 14, Table 2, Last Column*
> *The figure 13,33,000 makes no sense (even though it seems to be quoted directly from the Baluchistan Agency Adminstration Report on Figure 3). Does this mean 1,333,000 or 13,330,000 ? At any rate, if a proper rendition of this number is given, then an exchange rate to a more universal currency should be included (e.g., Rp. XXXXX, equivalent to present-day YYYYY £ or US $ ZZZZ or TTTT _).*

At that time the system used in the region would count as ten lac lac, ten thousand thousand, hundred ten unit so the figure 13,33,000 would be read as thirteen lac and thirty-three thousand. For the convenience of the readers the commas have been replaced to match the more renowned number system. The number now reads as **1,333,000**.
A column to the extreme right of the table has been added that shows the present-day equivalent of financial damages in US $.

> *The English of the paper should be improved throughout. There are articles, occasionally verbs, missing. Dr. Brian Atwater's name is misspelt in the Acknowledgments, etc.*

The revised manuscript will be checked for the language using a commercial software.

Spelling for Dr. Brian has been corrected.

---

## Author Comment (AC5)

**Response to reviews of RC2**

We are very thankful to the Dr. Issa El-Hussain for his constructive reviews that have improved the paper. Appended below, in italics and indented, are the full comments from RC2. Our responses follow each of the comments.

> *I would like to have seen a map showing the EQ location as well as the three town locations so that the reader see the relative distances between them.*

This map surely adds to the clarity for the readers. We will add it in the revised manuscript.

> *Figure 1, a, and b may need to redraw while keeping the original copy so that the reader would be able to see what is written in them, also translate c.*

Figures 1a and b have been resized for readability and a transliteration of figure 1c will be added to the revised manuscript.

> *Figure 3, authors may need to write the fuzzy words at the beginning of the paragraph to be able to understand the meaning (besides keeping the original).*

Point taken. Figure 3 will be improved for the revision.

> *The authors may need to say why the runup at Gwadar is very high relative to the wave height (ten times), is there energy focusing here? or eyewitness exaggeration? Or mixing with other flood events?*

We are extremely thankful to Dr. Issa El-Hussain for raising this question. The two dams (Mulla and Shadu band) were near the foot of Koh-e-Batil. After looking carefully at old map of Gwadar city, we realized that there was a road that led from the near coast area to the area where these dams are. But still, this could not lead the tsunami up to the dams. One possibility was the spill of water from the dams. Another possibility was misinterpretation of the eyewitness (Amina) account. Unfortunately, Amina has passed away and we could not re-interview her so we interviewed the interviewer and the interpreter to resolve this mystery. Upon interviewing we came to know that there are two old neighborhoods in Gwadar by the names of Mulla Band and Shadu Band which were the sites mentioned by Amina and not the exact locations of the dams. These new locations give runup elevations of 4 m (Mohallah Band, area adjacent to cricket stadium) and 5 m (Shadu Band, area adjacent to new football ground). Therefore, maximum runup at Gwadar turns out to be 11 m, that was for Jamat Khana.

---

## Author Response (AR1)

Revisions made in response to reviews of

**"Tsunami heights and limits in 1945 along the Makran coast estimated from testimony gathered seven decades later in Gwadar, Pasni and Ormara"**

a manuscript by Hira A. Lodhi, Shoaib Ahmed and Haider Hasan
intended for publication in Natural Hazards and Earth System Sciences (nhess-2021-53)

The revised manuscript contains minor changes that address the concerns of the two reviewers. Appended below, in italics and indented, are the full comments from both reviewers. Our responses follow each of the comments and new text added to manuscript is in bold. Revised manuscript being the file named <nhess-2021-53_Revision5.doc>

RESPONSE TO COMMENTS FROM RC1:

> *A major conclusion of this study seems to be that the number of fatalities was at most 150 (Table 2). This is in contrast to the figure of 4000 reported by the NOAA Tsunami Database.The discussion in the present paper would suggest populations of about 6000 in Gwadar (Line 72), 4000 in Pasni (Line 104, even though a newspaper reports 7000 people homeless) and perhaps 1000 in Ormora (Line 160), for a total of 11,000. The rest of the coast was probably very scarcely populated. A death toll of 4000 would amount to 1/3 of the total population, and would be an extremely high rate with long-lasting consequences on the economy of the province. It would probably have been mentioned repeatedly during the interviews of the (then very few) survivors. In this context, the NOAA figure is most probably grossly overestimated.*
> *\* Some discussion of this finding should be provided in the paper.*

[revised manuscript text omitted]

*The authors fail to mention the quantitative compilation carried out across the border in Iran by Okal et al. [2015].*

Point taken. This study is mentioned in the revised version, on Pg. 2, lines 50–53.
**A study by Okal et al., (2015), also based on field survey and eyewitness accounts quantizes the runup data along a 280 km long segment of Iranian shore. The study reports runup between 2.3–13.7 m and a time delay in the arrival of tsunami, indicating a secondary mechanism such as a landslide.**

*The authors mention Atwater et al. [2013] as a reference to tsunami surveys conducted for historical tsunamis many years after the event. However, this technique was pioneered a decade earlier for the 1946 Aleutian tsunami by Okal et al. [2002], which should probably be referenced.*

Reflected in the revised manuscript on Pg. 2, lines 45 & 46.
**However, this technique was pioneered by Okal et al., (2002) and was applied first for the Auletian tsunami.**

*The authors fail to reference the authoritative work of Ambraseys and Melville [1982] from which most of the information in Dominey-Howes et al. [2006] and Pararas-Carayannis [2006] is derived.*

Refence included Pg. 19, lines 271–272.

*Page 2, Line 54*
*The reference to Byrne and Davis [1992] should not include first names (by the way, Dr.*
*Byrne's is misspelt), and should really be Byrne et al. [1992] since the full authorship of*
*that paper includes Professor L.R. Sykes, whose name has been reduced to his initials*
*(L.R.S.) in the reference list.*

Corrected on Lines 57 and 60 also on line 233. The reference list has also been corrected.

*The coordinate scales on Figures 2, 5, 8 are completely out of range. Note that the longitude scales from 26°E through 176°E to 34°W. The latitudes are similarly extravagant.*
*\* The captions for these figures should name the specific cities.*

The figures have been corrected and replaced.

*There are some obvious discrepancies in the ages quoted for the witnesses. Not withstanding*

*the difficulty of obtaining their ages (as discussed, e.g., Okal et al. [2015], the latter should be consistent.*
*Note for example the case of Ms. Amina on Table 1. She is quoted as being $\geq 100$ yrs. old*
*at the time of the interview*
*(Note that 100+ is not a proper scientific notation. Use the symbols $>$, $\geq$, etc.)*
*but only 20 in 1945. She would then have been born in 1925, which would make her at*
*most 90 in 2015 or 95 in 2020.*
*Similarly, Ajyani Guli cannot have been 11 in 1945 (b. 1934) and already $\geq 90$ at the time of*
*the interview.*

*All information should be metric. Convert feet to meters throughout.*

Agreed that there are discrepancies in age. The ages in 1945 and at the time of interview, were quoted directly from "Remembering the 1945 Makran Tsunami; interviews with survivors beside the Arabian Sea". We should have been more skeptical towards the ages and should have discussed the discrepancies and the reasons for it in the paper. Table 1 has been updated to eliminate the column with ages at the time of interview.
The revised manuscript now uses metric system for the units throughout.

*Page 2, Line 32*
*The authors should emphasize the difference between the 2013 event for which a definitive*
*tsunami requiring a landslide was observed, and the landslide on the Owen Ridge*
*[Rodriguez et al., 2013] which is well documented, but for which the tsunami attacking*
*Oman can only be inferred.*

Change made.

*Page 2, Line 57*
*The earthquake was followed by five **recorded** aftershocks. There probably were many*
*more.*

Point taken and the word "recorded" added to the sentence on Page 2, Line 57.

*Section 3.1*
*There are references to Table 0.1) and Fig 0.2. This needs to be corrected.*

Corrected.

*Figure 1*

*Part (a) of the figure is hardly legible. I had to use a magnifying glass to decipher it. Translate the material in Arabic (or is it another language?) in Part (c), which will otherwise be completely useless to most of the readership.*

Figures have been revised to address the specific comments.

*Page 14, Table 2, Last Column*
*The figure 13,33,000 makes no sense (even though it seems to be quoted directly from the Baluchistan Agency Adminstration Report on Figure 3). Does this mean 1,333,000 or 13,330,000 ? At any rate, if a proper rendition of this number is given, then an exchange rate to a more universal currency should be included (e.g., Rp. XXXXX, equivalent to present-day YYYYY £ or US $ ZZZZ or TTTT _).*

At that time the system used in the region would count as ten lac lac, ten thousand thousand, hundred ten unit so the figure 13,33,000 would be read as thirteen lac and thirty-three thousand. For the convenience of the readers the commas have been replaced to match the more renowned number system. The number now reads as **1,333,000**.
A column to the extreme right of the table has been added that shows the present-day equivalent of financial damages in US $.

*The English of the paper should be improved throughout. There are articles, occasionally verbs, missing. Dr. Brian Atwater's name is misspelt in the Acknowledgments, etc.*
The revised manuscript has been checked for the language using a commercial software.
    Spelling for Dr. Brian has been corrected.

RESPONSE TO COMMENTS FROM RC2:

*I would like to have seen a map showing the EQ location as well as the three town locations so that the reader see the relative distances between them.*

An index map showing the three towns and epicenter location as reported by different studies has been added to the revised manuscript as Fig.1.

*Figure 1, a, and b may need to redraw while keeping the original copy so that the reader would be able to see what is written in them, also translate c.*

Figures (now Fig. 2a and 2b) have been resized for readability and a transliteration of figure 1c ( now Fig. 2c) has been added to the revised manuscript.

*Figure 3, authors may need to write the fuzzy words at the beginning of the paragraph to be able to understand the meaning (besides keeping the original).*

Point taken. Figure has been improved and includes a text box with the "*fuzzy text*" typed out.

*The authors may need to say why the runup at Gwadar is very high relative to the wave height (ten times), is there energy focusing here? or eyewitness exaggeration? Or mixing with other flood events?*

Upon interviewing we came to know that there are two old neighborhoods in Gwadar by the names of Mulla Band/ Mohalla Band and Shadu Band which were the sites mentioned by Amina and not the exact locations of the dams. These new locations give runup elevations of 6 m (Mohallah Band, area adjacent to cricket stadium) and 6 m (Shadu Band, area adjacent to new football ground). Therefore, maximum runup at Gwadar turns out to be 11 m, that was for Jamat Khana. The revised manuscript contains new estimates and text to describe this (Pg.17, lines 213–226) and a figure (Fig. 3) to reflect on this. The text adde to the manuscript is below:

**At Gwadar, although there was not much damage the maximum runup is found to be 11 m and the maximum inundation extent is around 900 m. These extents have been derived from the landmarks identified by the eyewitnesses but one of the eyewitnesses (Master Abdul Majeed) also reported, "Water came from the east and crossed to the other side" which is indicative of tsunami engulfing the entire landmass along the east to west stretch. None of the other eyewitnesses reported such inundation, The study does not use this account to conclude that the water might have swept across the entire tombolo as many other survivors had reported water reaching up to certain landmarks only. Another survivor of the event, Amina reported that the "huge wave" did not enter the city. She further reported the water reached the mosque; water was everywhere with no place to go but the water went further than the mosque. She also named some places that were inundated by the tsunami, such as the Mulla band and Shadu band (Kakar et al., 2015b). The water reaching the Mulla Band, reported by Amina and Hasan Ali might be that they were reporting "Mohalla Band" rather than "Mulla Band" or "Mohalla Band" is the new name of the neighbourhood just beside the Gwadar Miniport which was previously called as "Mulla Band", an area that is very likely to be inundated during the 1945 event. Shadu Band is another neighbourhood beside the new football stadium of Gwadar. In order to be sure if the interpretation of the locations was right, interviewers of the Amina were interviewed as Amina had passed away.**